# Effect of Oregano Oil and Cobalt Lactate on Sheep In Vitro Digestibility, Fermentation Characteristics and Rumen Microbial Community

**DOI:** 10.3390/ani12010118

**Published:** 2022-01-04

**Authors:** Zhengwen Wang, Xiongxiong Li, Lingyun Zhang, Jianping Wu, Shengguo Zhao, Ting Jiao

**Affiliations:** 1College of Animal Science and Technology, Gansu Agricultural University, Lanzhou 730070, China; gsauwangzhengwen@163.com (Z.W.); zhanglingyun1110@163.com (L.Z.); 2Key Laboratory of Grassland Ecosystem, Gansu Agricultural University, Lanzhou 730070, China; lxx_gsau@163.com; 3College of Grassland Science, Gansu Agricultural University, Lanzhou 730070, China; 4Gansu Academy of Agricultural Sciences, Lanzhou 730070, China; wujp@gsagr.ac.cn

**Keywords:** in vitro, oregano, essential oil, cobalt lactate, fermentation parameters, rumen microorganism

## Abstract

**Simple Summary:**

In the context of a shortage of feed resources and a complete ban on veterinary antibiotics, searching for green additives that can improve the production performance of ruminants has become a popular research topic. Oregano essential oil (EO) inhibits rumen gas production (GP) and regulates animal digestive metabolism, and cobalt lactate (Co) can improve feed digestibility. However, previous studies on EO of oregano and Co showed different results. Therefore, the present study aimed to investigate the effects of different EOC addition levels on rumen in vitro fermentation and rumen bacterial community composition, and the experimental data obtained showed that all five EOC (0.1425% cobalt lactate + 1.13% oregano essential oil + 98.7275% carrier) addition levels in this experiment had no significant effect on nutrient digestibility. However, the addition of 1500 mg·L^−1^ EOC significantly improved rumen fermentation parameters and altered the microbiota composition. All presented data provide a theoretical basis for the application of oregano essential oil and cobalt in ruminant nutrition.

**Abstract:**

The objective of this experiment was to evaluate the effect of different EOC (0.1425% cobalt lactate + 1.13% oregano essential oil + 98.7275% carrier) levels on in vitro rumen fermentation and microbial changes. Six EOC levels (treatments: 0 mg·L^−1^, CON; 50 mg·L^−1^, EOC1; 100 mg·L^−1^, EOC2; 400 mg·L^−1^, EOC3; 800 mg·L^−1^, EOC4 and 1500 mg·L^−1^, EOC5) were selected to be used to in vitro incubation. The in vitro dry matter digestibility (IVDMD), in vitro neutral detergent fiber digestibility (IVNDFD), in vitro acid detergent fiber digestibility (IVNDFD), pH, ammonia-nitrogen (NH_3_-N) concentration, total volatile fatty acid (TVFA) concentration and microbial protein (MCP) concentration were measured after 48 h incubation, after which the groups with significant nutrient digestibility and fermentation parameters were subjected to 16S rRNA sequencing. The results showed that the total gas production (GP) of the EOC5 group was higher than that of the other groups after 12 h of in vitro incubation. TVFA, NH_3_-N and MCP concentrations were also shown to be higher in group EOC5 than those in other groups (*p* < 0.05), while NH_3_-N and MCP concentrations in the EOC2 group were lower than those in other groups significantly (*p* < 0.05). The molar ratio of acetic acid decreased while the molar ratio of propionic acid increased after the addition of EOC. 16S rRNA sequencing revealed that the rumen microbiota was altered in response to adding EOC, especially for the EOC5 treatment, with firmicutes shown to be the most abundant (43.1%). The relative abundance of *Rikenellaceae_RC9_gut_group* was significantly lower, while the relative abundance of *uncultured_bacterium_f_Muribaculaceae* and *Succiniclasticum* was significantly higher in the EOC5 group than those in other groups (*p* < 0.05). Comprehensive analysis showed that EOC (1500 mg·L^−1^) could significantly increase gas production, alter sheep rumen fermentation parameters and microbiota composition.

## 1. Introduction

The exploitation and utilization of straw feed has always been an important subject in animal nutrition, as massive burning leads to serious waste of crop straw, although it is difficult to apply to monogastric animals [1]. However, the microbial community of ruminants ensures that they can produce high-quality proteins from low-quality plant feed and can catalyze the hydrolysis of indigestible plant crude fibers through microbial extracellular enzymes [2]. Although the rumen microbial community is very stable throughout the life cycle of animals, the diversity of the microbial community is largely influenced by diet (e.g., different feed types and additives) [3]. Some studies have concluded that supplementing feed with additives such as probiotics has a positive effect on the gastrointestinal microbiota of cattle [4]. Therefore, the rational use of feed additives to improve the gastrointestinal flora of ruminants is highly important for improving the utilization of feed resources.

The use of antibiotics in animal feed has been almost banned in the European Union since 2006 [5]. Similarly, China’s Ministry of Agriculture and Rural Affairs issued Announcement No. 194 in 2020, which completely banned the addition of antibiotics to feed. Therefore, the identification of green additives for ruminant productivity has become a popular research topic. Oregano essential oil (EO) is an aromatic volatile oil extracted from oregano and has strong antibacterial and antioxidant properties [6] that help maintain a healthy microbiota by regulating the proliferation of beneficial microorganisms in the gastrointestinal tract and suppressing increases in the number of harmful microorganisms [7]. Previous studies have reported that EO inhibits rumen gas production (GP) [8], and more recent studies have found that EO regulates animal digestive metabolism by improving protein metabolism and volatile fatty acid (VFA) patterns [9]. Claudia found that EOs improve rumen fermentation and feed digestion efficiency by affecting bacteria of the families *Prevotellaceae*, *Lachnospiraceae* and *Ruminococcaceae* [10]. Vitamin B12 acts as an activator of a variety of enzymes, and its synthesis requires the participation of cobalt (Co) [11], and Co has been shown to improve feed digestibility in both in vitro and in vivo studies [12,13].

Previous studies have shown that calf growth is improved when oregano EO and Co combined (EOC) are added to milk given to calves [14]. Moreover, dietary EOC supplementation significantly increased average daily weight gain in goats while significantly improving carcass weight and meat quality, and EOC was found to stimulate changes in the animals’ immune system at the physiological and cellular levels [15]. These experiments showed that providing a mixture of Co and EO improved rumen fermentation capacity and altered VFA patterns, resulting in improved digestibility of dry matter, protein and fiber. However, previous studies on oregano EO and Co have yielded inconsistent results on the basis of the amount and type of EO additive, and few studies have reported relationships between rumen microbes and rumen fermentation parameters and feed digestibility [16]. This study hypothesized that different levels of EOC addition would have different effects on rumen microorganisms, while nutrient digestibility and fermentation parameters might change with the microbiota altered. Therefore, the purpose of this experiment is to evaluate the effects of different EOC addition levels on in vitro fermentation, nutrient digestibility and rumen bacterial community composition by comprehensive analysis of rumen microbes, nutrient digestibility and in vitro rumen fermentation parameters, and determine the optimal addition level.

## 2. Materials and Methods

### 2.1. Feeding and Management

All experiments involving animals were reviewed and approved by the Animal Committee of Gansu Agricultural University (GSAU-2th-AST-2021-126). 

Five healthy small-tailed han sheep (approximately 10 months of age) with permanent rumen fistulae were used as test animals; they had a mean rectal temperature of 39 °C and a mean body weight of 50 kg. The sheep were fed twice daily (9:00 a.m. and 5:00 p.m.) in a single barn and had free access to water. The composition of the ration is shown in Table 1. According to the agricultural industry standard for sheep feeding (NY/T816-2004), a basic diet with a daily feed intake of 3.06 kg·d^−1^ was prepared (Table 1).

### 2.2. Oregano Essential Oil Organic Cobalt Treatment

EOC (0.1425% cobalt lactate + 1.13% oregano essential oil + 98.7275% carrier) was purchased from Ralco Animal Nutrition (Marshall, MN, USA). The carrier was composed of 75% zeolite (mordenite) + 15% limestone + 10% diatomite. Previous studies by our group have demonstrated that this carrier has no effect on sheep [17]. Six EOC inclusion levels were evaluated: 0 mg·L^−1^ (CON), 50 mg·L^−1^ (EOC1), 100 mg·L^−1^ (EOC2), 400 mg·L^−1^ (EOC3), 800 mg·L^−1^ (EOC4) and 1500 mg·L^−1^ (EOC5). These levels were adjusted to compensate for the amount of EOC in the dry granular product added to the in vitro fermentation medium. 

### 2.3. In Vitro Fermentation

Rumen contents were obtained from 5 sheep at the same ratio before morning feeding. After mixing, the mixture was filtered through a preheated thermoflask with 4 layers of gauze, CO_2_ was continuously injected, and the flask was set aside. The same feed as that provided to the test sheep (0.2000 ± 0.0010 g, dry matter basis) was accurately weighed into a nylon bag with a pore size of 50 μm, a length of 2.0 cm and a width of 3.0 cm. It was placed at the bottom of the extracorporeal tracheal tube, 30 mL microbial culture solution (10 mL rumen juice + 20 mL buffer) was added and it was always filled with CO_2_ gas; the buffer was prepared according to Menke [18]). After the gas was discharged from the gas-producing tube, the rubber sleeve at the front of the gas-producing tube was immediately sealed, and the scale value (mL) of the gas-producing tube was recorded. The GP volume readings (mL) were recorded at 2, 4, 6, 9, 12, 24, 36 and 48 h on a thermostatic water bath stand at 39 °C, and each reading was followed by a slight vibration to simulate rumen movement. The test tube was a 100 mL glass syringe (Germany). There were 6 replicates per treatment with 6 blank vials (rumen fluid only) without feed substrate to correct for background GP.

### 2.4. Sampling

After 48 h of incubation, the nylon bags were removed and put in ice water to stop fermentation. The pH of the culture broth was measured immediately, and the nylon bags were washed with distilled water until they became colorless. The culture solution was put into 5 mL centrifuge tubes and centrifuged at 3000 r·min^−1^ for 10 min. The supernatant was separated and stored at −20 °C, the concentrations of VFA, acetic acid, propionic acid, butyric acid and NH_3_-N were determined; the remaining part was stored at −80 °C for analysis of microbial protein (MCP) and bacterial community composition.

### 2.5. DNA Extraction and High-Throughput Sequencing

Rumen microbiome DNA was extracted using a bacterial DNA extraction kit (Omega, Shanghai, China), fresh samples of 250–500 mg were transferred into 2.0 mL centrifuge tubes (grinding beads were added), SL2 was added at 700 uL (SL2 is preferred for a first extraction; if poor, SL1 is used for reextraction), 150 uL of Enhancer SX was added, the tube was capped tightly and vortexed to mix thoroughly. Then the sample was lysed, precipitated to remove impurities, filtered to remove inhibitors and finally the DNA was eluted. Finally, the concentration and purity were checked using an ultramicrospectrophotometer (Thermo NanoDrop 2000, Thermo Fisher, Shanghai, China). The extracted DNA samples were stored at −80 °C. The V3-V4 region of the 16S rRNA gene was amplified by PCR using universal primers 338 F (5’-ACTCCTAC GGGAGGCAGCAG-3’) and 806 R (5’-GGACTACHVGGGTWTCTAAT-3’) to characterize the rumen microbiota, and PCR amplification was performed as follows: pre-denaturation for 3 min (95 °C); denaturation for 30 s (95 °C), annealing for 30 s (55 °C), extension for 30 s (72 °C), 40 cycles; final extension for 7 min (72 °C). All the amplification products were analyzed by library sequencing on the Illumina MiSeq (Illumina, San Diego, CA, USA) platform.

### 2.6. Bioinformatic Analysis

Based on the raw data returned from the Illumina HiSeq sequencing platform, optimized sequences (tags) were obtained by double-end splicing (FLASH v 1.2.7), filtering (Trimmomatic v 0.33), and chimera removal (UCHIME v 4.2). Tags were clustered at a 97% similarity level, operational taxonomic units (OTUs) were obtained using USEARCH software and the OTUs were taxonomically annotated based on the Silva (bacteria) taxonomic database. Based on the results of the OTU analysis, the data were analyzed to generate community structure maps, species clustering heat maps, phylogenetic trees and taxonomic dendrograms of each sample at the phylum, class, order, family, genus and species levels. The QIIME2 software was used to calculate the ACE, Chao1, Shannon and Simpson alpha diversity indices to analyze the species diversity within individual samples. Sample dilution curves and rank abundance curves were plotted. Beta diversity analysis was used to assess the magnitude of differences between samples in terms of species diversity (colony composition and structure). Sample hierarchical clustering (unweighted pair group method with arithmetic mean (UPGMA)), nonmetric multidimensional scaling (NMDS) analysis, sample clustering using heat maps, sample principal component analysis (PCA) and principal coordinate analysis (PCoA) (with grouping information) were performed at the corresponding distances according to the distance matrix. To identify biomarkers with significant differences between groups through linear discriminant analysis effect size (LEfSe), the species abundance data were compared between groups using t-tests in Metastats software, after which the data were screened for species that caused differences in sample composition between the two groups.

### 2.7. Chemical Analysis

The in vitro dry matter digestibility (IVDMD) was determined by drying at 65 °C for 48 h (AOAC, 2016; 930.15), in vitro neutral detergent fiber (AOAC, 2016; 973.18) and in vitro acid detergent fiber (AOAC, 2016; 2002.04) using the AOAC international procedures [19]. A pHS-3C acidometer was used to measure the pH, and the NH_3_-N concentration was determined by the phenol-sodium hypochlorite colorimetric method according to Wang et al. [20]. VFAs were determined by gas chromatography (GC-2010 Plus; Shimadzu, Kyoto, Japan) according to the method of Zhou et al. [21]. Using internal standards, internally labeled 2-ethylbutyric acid (2 EB) was used. A 30 m × 0.32 mm× 0.25 μm size capillary column (AT-FFAP) was used. The Gas chromatograph column temperature was programmed for 1-minute hold (60 °C), 5 °C·min^−1^ to 115 °C (not held), 15 °C·min^−1^ to 180 °C. Detector temperature was set to 260 °C and injection port temperature to 250 °C. MCP concentrations were measured using a kit (Nanjing Jiancheng Institute of Biological Engineering) with a microplate reader (Thermo Fisher Scientific).

*GP* was measured according to the following formula:$$GPt=200\times \left(Vt-Vo\right)/W$$
where, *t* is a certain time point recorded during fermentation (h); *GPt* is the output tracheal reading of each combination at time *t* (mL); *V0* is the 0 h reading of each producing pipe (mL); *Vt* is the reading of producing trachea after t h of in vitro culture (mL) and *W* is the dry matter weight (mg) of each combination sample in the production pipe. *GP* (mL) at a specific point in time = *GP* of the sample during this period—Initial reading of the producing pipe—*GP* of the blank tube during the period.

### 2.8. Statistical Analysis

We used a one-way experimental design. Before any statistical analyses were conducted, we used SPSS (SPSS v 26.0, SPSS, Inc., Chicago, IL, USA)’s Kolmogorov–Smirnov test program to check the normality and outliers of all data. Levene’s test was used to test the homogeneity of variance, and the experimental data were analyzed by one-way ANOVA (Analysis of variance) of SPSS (SPSS v 26.0, SPSS, Inc.) software. Duncan’s method was used for multiple comparisons when the difference was significant, *p* < 0.05 was considered statistically significant and 0.05 *< p <* 0.10 was considered a trend towards significant difference. The correlations between nutrient digestibility, in vitro fermentation parameters and rumen microbes (relative abundance > 0.5%) were analyzed by Spearman correlation tests.

## 3. Results

### 3.1. Effect of In Vitro Fermentation Total Gas Production by EOC

Within 12 h, the GP in the CON group was higher than that in the other groups, and after 12 h, the GP in the EOC5 group was higher than that in the other groups (Figure 1).

### 3.2. Changes in Feed Digestibility and Fermentation Parameters

As shown in Table 2, for GP_24 h_ the EOC5 group was significantly higher than that in the CON group (*p* < 0.05), and that in the CON group was significantly higher than that in all the other groups (*p* < 0.05). The IVDMD, IVNDFD and IVADFD did not differ significantly among groups (*p* > 0.05).

As shown in Table 3, the of total volatile fatty acids (TVFAs), NH_3_-N and MCP concentrations in the EOC5 group were significantly higher than those in all the other groups (*p* < 0.05), the NH_3_-N and MCP concentrations in the EOC2 group were significantly lower than those in all the other groups (*p* < 0.05), and the TVFA concentration in the EOC1–EOC4 groups was significantly lower than that in the CON group (*p* < 0.05). In terms of the proportion of various VFAs, propionate was found to be the lowest in the CON group (*p* < 0.05), acetate and isovaleric acid and the ratio of acetic acid to propionic acid (A:P) were significantly higher in the CON group than in all the other groups (*p* < 0.05), butyrate was the highest in the EOC5 group (*p* < 0.05), valerianic acid was highest in the EOC1 group (*p* < 0.05) and the pH was not significantly different among the groups (*p* > 0.05).

### 3.3. Alteration in the Composition of the Rumen Bacterial Community

A total of 1,427,227 pairs of reads were obtained by high-throughput sequencing of the CON, EOC2 and EOC5 groups. A total of 1,413,142 clean tags were generated after splicing and filtering of the double-ended reads, with at least 71,328 clean tags generated for each sample and an average of 78,508 clean tags with an average sequence length of 420 bp. The number of OTUs for each sample was obtained by clustering tags at the 97% similarity level using USEARCH software, and a total of 597 OTUs were obtained, with 589 OTUs in the CON group, 584 OTUs in the EOC2 group and 596 OTUs in the EOC5 group (Figure 2A). The dilution curve depicted the species diversity and species richness of each sample, and the curves plateaued at 20,000 reads, indicating that the sequencing coverage was saturated (Figure 2B). The alpha diversity index has no significant difference (Table 4). A graph of the species accumulation shows that the number of species and shared species in the environment reached saturation with increasing sample size and did not increase with increasing sample size. NMDS (Figure 2C) showed that there were significant differences among the three groups of microbial species, and ANOSIM analysis further showed that the between-group differences were significantly greater than the within-group differences (Figure 2D).

### 3.4. Alteration of Rumen Microbial Composition Caused from EOC

At the taxonomic level, 18 phyla, 25 classes, 32 orders, 62 familys, 164 genus and 194 species were detected. At the phylum level, Bacteroidetes, Firmicutes, Proteobacteria and Synergistetes were the dominant phyla, with a relative abundance greater than 1%, and the relative abundance of Bacteroidetes and Firmicutes was the highest among all the groups, accounting for more than 88% of the total abundance (Figure 3A). The Bacteroidetes content was highest in the CON group (49.2%), lowest in the EOC5 group (46.8%) and highest in the EOC5 group (43.1%). At the genus level (Figure 3B), 73 genera had a relative abundance greater than 0.1%. *Uncultured_bacterium_f_Muribaculaceae*, *Rikenellaceae_RC9_gut_group* and *uncultured_bacterium_f_F082* were the dominant genera in all three groups, where the top 10 species in terms of abundance of *Rikenellaceae_RC9_gut_group* had significantly lower relative abundance in the EOC5 group (*p* < 0.05), *uncultured_bacterium_f_F082* had significantly lower relative abundance in both the EOC5 and EOC2 groups than in the CON group (*p* < 0.05), *uncultured_ bacterium_f_Muribaculaceae* and *Succiniclasticum* were significantly higher in terms of relative abundance in the EOC5 group (*p* < 0.05) and the abundance of the *Lachnospiraceae_NK3A20_group* was the lowest in the CON group, but the difference was not significant. ANOVA revealed significant differences in microflora between groups at the phylum and genus levels. LEfSe of samples between groups revealed (Figure 4) that there were 9, 7 and 3 biomarkers that significantly differed (LDA score > 4) in the CON, EOC2 and EOC5 groups, respectively, similar to the results of ANOVA, with significant differences between groups.

### 3.5. Interactions among In Vitro Digestibility, Fermentation Parameters and Rumen Microorganisms

A correlation heat map between microbial bacteria at the genus level in vitro in the rumen (relative abundance > 0.5%) and nutrient digestibility and fermentation parameters (correlation threshold > 0.1) was constructed, which is shown in Figure 5. DMD, NDFD and ADFD were found to cluster together, and GP and in vitro fermentation parameters clustered together. A total of 32 microbes at the genus level were significantly correlated, of which 16 were significantly positively correlated and of which 16 were significantly negatively correlated (*p* < 0.05). The *uncultured_bacterium_f_Muribaculaceae*, *Succiniclasticum*, *Saccharofermentans*, *Lachnoclostridium_1 and Family_XIII_AD3011_groups* were significantly positively correlated with GP and fermentation parameters (*p* < 0.05), and bacteria such as *Rikenellaceae_RC9*, *Prevotellaceae_UCG-001* and *Prevotellaceae_UCG-003* showed significant negative correlations (*p* < 0.05) with in vitro fermentation parameters.

## 4. Discussion

Previous studies had shown different effects of oregano essential oil and cobalt on rumen microbial activity and fermentation characteristics in ruminants due to different additive types and adding levels [16], which may be related to the type of EO, its structure and the substrate employed. Therefore, to develop new additives for EO binding to ultimately effectively regulate ruminant production, in this study, the effects of different gradients of oregano essential oil and cobalt lactate binding on nutrient digestibility, rumen fermentation patterns and microbiota were evaluated by in vitro incubation.

GP is an important indicator for feed fermentability and rumen microbes activities [22]. Methane accounts for a large portion of total rumen gas production in vitro. Macheboeuf found that 5 mM oregano EO inhibited 98% of methane production [23], because the phenolics in EO inhibit methanogenic and hydrogen-producing bacteria from reducing methane production rapidly [24]. So EOC led to the reduction of total GP at 12 h by inhibiting methane production in this study. The inhibition of EO on methanogenic bacteria was greater than that of other bacteria [25], which was one of reasons why we did not observe significant changes in the digestibility of nutrients. Adding cobalt to the diet could accelerate the rate of the rumen flora to synthesize vitamin B12 [26]. The high gas production in the EOC5 group after 12 h was due to the involvement of vitamin B12 as a coenzyme factor in the metabolism of carbohydrates to produce large amounts of carbon dioxide [27]. Nutrient digestibility reflects the degree of degradation of the feed by microbes in a fermentation system and the ease of digestion and utilization by animals [28]. Complex and diverse active chemical composition of plant essential oils results in different functional properties and effects on in vitro nutrient digestibility [29]. The addition of EOC below 800 mg·L^−1^ did not affect the nutrient digestibility while reducing VFA, because EO could reduce VFA by inhibiting starch hydrolyzing bacteria and protein hydrolyzing bacteria, but did not affect the activity of fiber degrading bacteria [30]. Co could form cross-linkages between bacteria and feed with negative electrical charge to improve feed digestibility [31], 1500 mg·L^−1^ EOC level is consistent with the inclusion levels that cobalt can quickly alter the concentration of rumen microbes and therefore promote rumen fermentation and nutrient degradation rates [26]. This suggests that EO and Co in EOC can produce synergistic effects in terms of mitigating methane emissions without reducing feed digestibility, increasing rumen cellulolytic bacteria activity and promoting rumen fermentation function.

Ruminal pH is an important indicator of the homeostasis of the rumen environment in ruminants and usually varies from 5.0 to 7.5 [32]. In this study, ruminal pH was at a normal level of 6.3–6.45. VFAs are the main products of carbohydrate degradation of food rations and provide most of the energy precursors for metabolic processes in ruminants. EO can reduce VFA production by inhibiting starch-hydrolyzing and protein-hydrolyzing bacteria [30], besides EO can reduce protozoa population and decrease acetic acid and total VFA [33]. The abovementioned reasons resulted in a significant reduction of TVFA and acetic acid when the EOC supplemental level was lower than 800 mg·L^−1^. However, we found that the molar ratio of propionic acid in all groups was significantly increased. Poudel found that adding EO can increase the concentration of propionic acid in the rumen of Holstein calves by increasing the content of *Prevotellaceae* [34]. As mentioned before, some microorganisms can use Co to synthesize vitamin B12 and other rumen microbial growth factors, and vitamin B12 deficiency can impair methylmalonyl CoA mutase activity, thus affecting propionate production and causing dysregulation of glucose homeostasis in ruminants [26], Co can promote nonstructural carbohydrate digestion and increase propionate ratios by reducing rumenomonas [35]. Higher propionic acid levels were generally considered beneficial for ruminant production [36]. The high concentration of EOC also increased the production of butyric acid in this study, which is generally considered to be a better VFA because it participates in the development of rumen papilla by stimulating rumen epithelial metabolism [34]. In summary, although the effects of different concentrations of EOC on the total TVFA are not consistent, this may be because the concentration thresholds for the best effects of oregano essential oil and cobalt lactate are not consistent. However, all gradients of EOC reduced the ratio of acetic acid to propionic acid (A:P) in the structure of VFA, which can increase the energy utilization of ruminants [37], and increase the molar ratio of beneficial VFAs such as propionic acid and butyric acid. Our research results show that EOC creates a VFA structure that is conducive to animal production. The concentration of NH_3_-N can reflect the degradation of nitrogenous substances in the diet and the utilization of ammonia by microbes, and some NH_3_-N can be synthesized by rumen microorganisms to synthesize MPC. Zhou reported that 92 mg·L^−1^ EO inclusion level reduced NH_3_-N concentration, we considered that EO reduces rumen ammonia nitrogen concentration and protein deamination by inhibiting high ammonia-producing bacteria [24]. The above evidence shows that EO at a level of about 100 mg·L^−1^ has the most effective inhibitory effect on NH_3_-N, while 1500 mg·L^−1^ EOC significantly increased NH_3_-N and MCP production. Our previous study found that the addition of 7 g/d of Co significantly increased NH_3_-N concentration in sheep [17], because high levels Co play a greater role in NH_3_-N synthesis and metabolism of nitrogenous substances [38]. The above results indicate that although EOC at different addition levels had different effects on a single index due to the two having different thresholds for optimal performance, overall oregano essential oil and cobalt exerted synergistic effects in changing rumen fermentation parameters such as VFA structure, NH_3_-N and MCP production, while the particular synergistic mechanisms and thresholds for both to exert their optimal effects remain to be studied in our subsequent studies.

After comparing the nutrient digestibility and fermentation parameters, we performed 16S rRNA sequencing. It was found that there was no significant difference in Alpha diversity. However, further analysis of NMDS and Anosim revealed that different EOC additions resulted in significant changes in the community composition and relative abundance of specific microorganisms, which may be due to the diversity of active ingredients in EO and the different sensitivity of different microorganisms to EO levels, resulting in differences in rumen microbial populations [39]. At the phylum level, Bacteroidetes and Firmicutes are the main dominant phyla. Firmicutes could improve the abundance of genes encoding enzymes related to energy metabolism [40], the high proportion (43.1%) of Firmicutes in the EOC5 group could promote the digestibility of oligosaccharides, starch and cellulose [41], then improve rumen fermentation level and help sheep digest and absorb nutrients. At the genus level, *uncultured_bacterium_f_Muribaculaceae, Rikenellaceae_RC9_gut_group* and *uncultured_bacterium_f_F082* were the dominant genus. *Rikenellaceae_RC9_gut_group* was positively correlated abnormal glucose and lipid metabolism with a high-fat diet [28]. After reducing the abundance of *Rikenellaceae_RC9_gut_group* and *Prevotellaceae_UCG-001*, the microorganisms that produced propionic acid and butyric acid increased due to the increase in available substrates [42]. We found that *Rikenellaceae_RC9_gut_group* was negatively correlated with in vitro fermentation parameters through correlation analysis, which showed that EOC changed the VFA composition by affected the abundance of *Rikenellaceae_RC9_gut_group*, and had a positive effect on maintaining the normal glucose and lipid metabolism of sheep. We also found that *uncultured_bacterium_f_Muribaculaceae* and *Succiniclasticum* exhibited high abundance in the EOC5 group, *uncultured_bacterium_f_Muribaculaceae* could produce succinic acid through degradation of polysaccharides [43]. As an important intermediate in the synthesis of propionic acid, succinic acid generates glucose by the gluconeogenic pathway [44], while *Succiniclasticum* could use succinic acid to produce propionic acid [45]. *uncultured_bacterium_f_Muribaculaceae* and *Succiniclasticum* were positively correlated with in vitro fermentation parameters in the present study, and, based on this, we hypothesized that the additive level of EOC5 was just suitable for the proliferation of this type of rumen bacteria that could produce propionic acid [46], and then increased the feed digestion rate and rumen fermentation. *Prevotellaceae_UCG-001* and *Prevotellaceae_UCG-003* abundance were significantly negatively correlated with fermentation parameters, which is consistent with Yang’s finding of a negative correlation between *Prevotellaceae_UCG-001* and *Prevotellaceae_UCG-003* and VFA concentration and NH_3_-N concentration [47]. Prevotellaceae belongs to the Bacteroidetes, probably due to the addition of oregano essential oil and cobalt lactate, which affect Firmicutes abundance, thus limiting the reproduction of these low-abundance species. Due to the complex composition of EOC, its antibacterial mechanism against a single strain was difficult to predict. It is reported that some small molecules contained in EO could also interact with the cell membranes of gram-negative bacteria, thereby reducing their specificity and regulating rumen fermentation [48]. Vitamin B12 could affect the abundance of Prevotella, Bacteroides, Rumenobacter, and D-shaped bacteria Vibrio and Vibrio succinate [49]. In this study, the results of EOC regulation of some rumen microflora were found, and the corresponding findings were that they changed gas production and rumen fermentation, reflecting the effects of these microorganisms on rumen fermentation and nutrient digestibility. However, this study lacks the effect of single EO and CO at different addition level on these factors, so the specific mechanisms of EOC effects on digestive metabolism and rumen microbiota in sheep are to be studied in our subsequent continuation.

## 5. Conclusions

The addition of 1500 mg·L^−1^ EOC significantly improved GP parameters, and rumen fermentation parameters and altered the microbiota composition. Although we are not sure about the concrete contribution of a single EO and Co on these effects, the addition of different concentrations of EOC to sheep rumen in vitro cultures increased the nutrient digestibility, and altered rumen fermentation to a mode in which acetic acid decreased and propionic acid increased.

## Figures and Tables

**Figure 1 animals-12-00118-f001:**
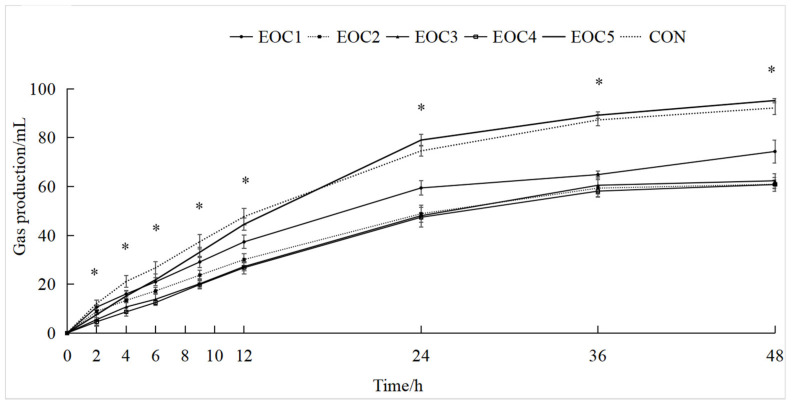
Variation of gas production at 0~48 h. * *p* < 0.05.

**Figure 2 animals-12-00118-f002:**
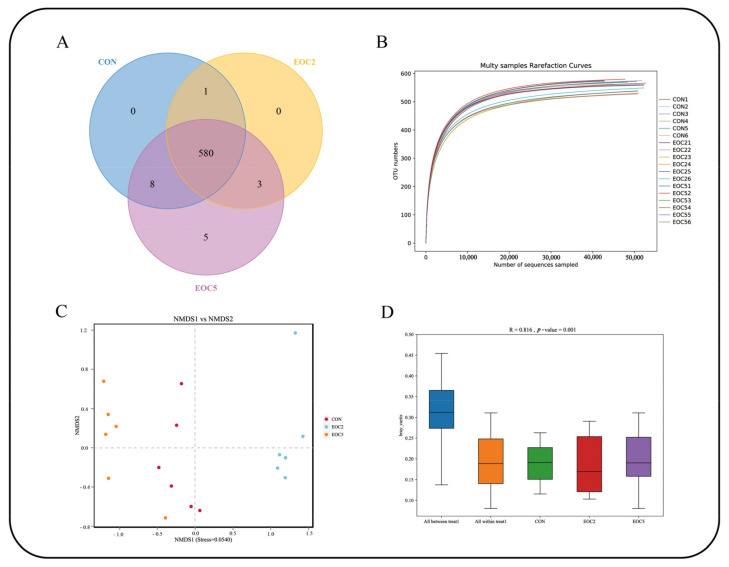
Diversity analysis. (**A**) operational taxonomic units-Venn (OTU-Venn) analysis; (**B**) dilution curve analysis; (**C**) nonmetric multidimensional scaling (NMDS) analysis; (**D**) Anosim analysis box plot.EOC, 0.1425% cobalt lactate + 1.13% oregano essential oil + 98.7275% carrier; CON, control group, that is, 0 mg·L^−1^ EOC addition level.

**Figure 3 animals-12-00118-f003:**
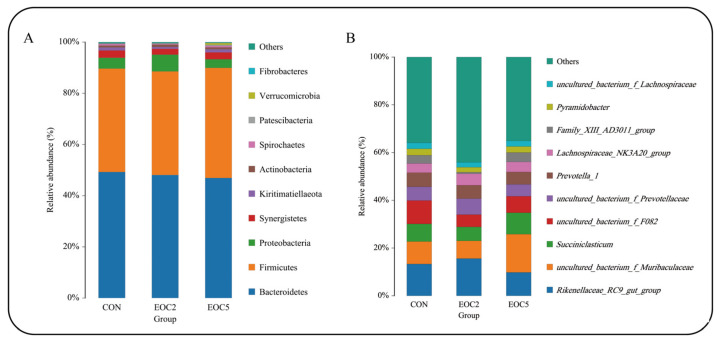
(**A**) Relative abundance of phylum horizontal species; (**B**) relative abundance of genus horizontal species.

**Figure 4 animals-12-00118-f004:**
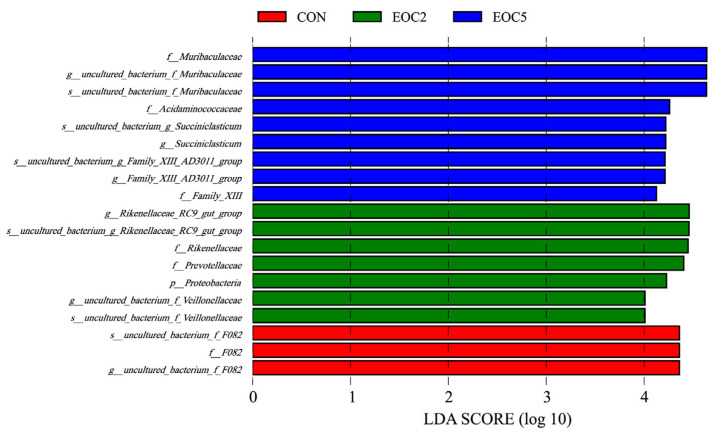
Significantly different bacterial taxa identified by the linear discriminant analysis effect size (LEfSe).

**Figure 5 animals-12-00118-f005:**
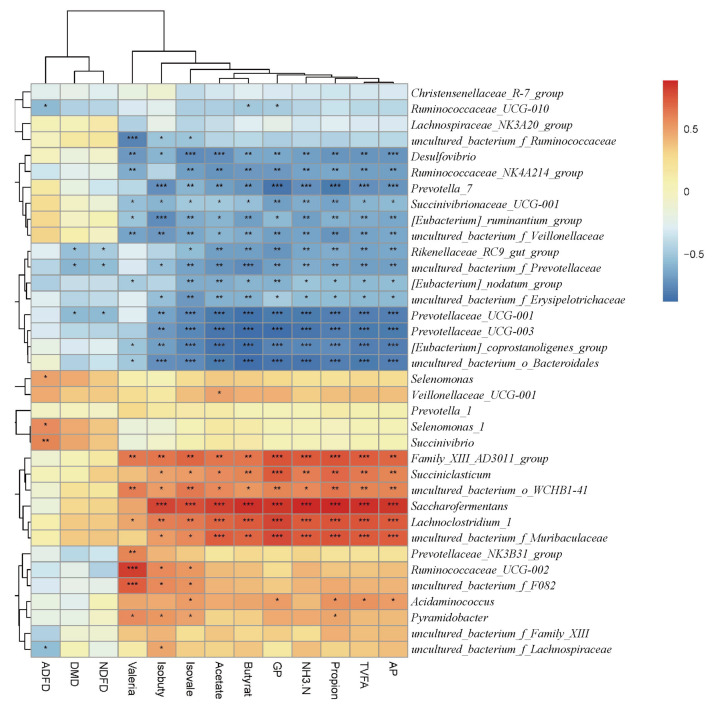
Correlation heat map. Note: ADFD, acid detergent fiber digestibility; DMD, dry matter digestibility; NDFD, neutral detergent fiber digestibility; GP, gas production; TVFA, total volatile fatty acids; AP, the ratio of acetic acid to propionic acid; * *p* < 0.05, ** *p* < 0.01, *** *p* < 0.001.

**Table 1 animals-12-00118-t001:** Composition and nutrient levels of the fully mixed diet (air drying basis).

Ingredients	Content (%)	Nutrient	Content
Whole corn silage	72.96	Digestible energy, ME/(M J kg^−1^)	11.58
Corn	14.41	Digestible energy, DE/(M J kg^−1^)	14.12
Wheat bran	4.00	Dry matter, DM (%)	36.23
Cottonseed meal	3.86	Crude Protein, CP (%)	14.73
Rapeseed meal	3.59	Calcium, Ca (%)	0.76
Calcium Hydrogen Phosohate	0.31	Phosphorus P (%)	0.65
Limestone	0.31	Ether extract, EE (%)	2.50
Salt	0.27	Neutral detergent fiber, NDF (%)	35.81
1% Premix ^1^	0.29	Acid detergent fiber, ADF (%)	20.01
Total	100		

^1^ One kilogram of the premix contained the following: VA 650,000 IU, VD_3_ 300,000 IU, VE 400,000 IU, Fe 500 mg, Cu 500 mg, Mn 1000 mg, Zn 500 mg, Co 15 mg, Se 40 mg. Metabolic energy and digestible energy were calculated, and other values are measured.

**Table 2 animals-12-00118-t002:** Effects of EOC on gas production parameters.

Items	EOC1	EOC2	EOC3	EOC4	EOC5	CON	SEM	*p*-Valve
GP_24 h_ ^1^ (mL)	59.4 ^c^	48.83 ^d^	48.03 ^d^	47.4 ^d^	79.00 ^a^	74.58 ^b^	2.79	<0.001
IVDMD ^2^ (%)	67.99	68.02	67.98	68.02	69.44	67.3	1.14	0.062
IVNDFD ^3^ (%)	59.27	58.8	57.98	59.68	60.79	58.4	2.06	0.21
IVADFD ^4^ (%)	35.76	34.94	33.07	35.49	34.5	32.89	3.57	0.654

^1^ GP_24 h_, 24-h gas production capacity. ^2^ IVDMD, in vitro dry matter digestibility. ^3^ IVNDFD, in vitro neutral detergent fiber digestibility. ^4^ IVADFD, in vitro acid detergent fiber digestibility. SEM: Standard error of the mean. ^a,b,c,d^ Values with different superscripts in the same row are significant different (*p* < 0.05).

**Table 3 animals-12-00118-t003:** Effects of EOC on rumen fermentation parameters.

Items	EOC1	EOC2	EOC3	EOC4	EOC5	CON	SEM	*p*-Valve
NH_3_-N (mg dL^−1^)	27.42 ^b^	24.76 ^c^	27.33 ^b^	27.32 ^b^	28.84 ^a^	27.04 ^b^	1.53	<0.001
TVFA (mmol l^−1^) ^1^	78.44 ^d^	68.89 ^e^	84.63 ^c^	77.01 ^d^	104.56 ^a^	95.55 ^b^	3.78	<0.001
MCP (mg mL^−1^) ^2^	3.48 ^b^	2.16 ^c^	3.59 ^b^	3.43 ^b^	3.91 ^a^	3.58 ^b^	0.16	<0.001
pH	6.41	6.42	6.38	6.45	6.30	6.35	0.17	0.313
Acetate (% ^3^)	46.1 ^c^	46.39 ^c^	47.77 ^b^	48.02 ^b^	48.53 ^b^	50.97 ^a^	0.81	<0.001
Propionate (%)	31.89 ^b^	34.01 ^a^	33.75 ^a^	32.46 ^b^	29.76 ^c^	28.75 ^d^	0.68	<0.001
Isobutyric acid (%)	1.98	1.88	1.89	1.84	1.99	1.96	0.15	0.008
Butyrate (%)	11.66 ^b^	10.88 ^c^	10.88 ^c^	11.69 ^b^	12.65 ^a^	10.59 ^c^	0.52	<0.001
Isovaleric acid (%)	3.84 ^a^	3.26 ^b^	3.03 ^b^	3.2 ^b^	4.04 ^a^	3.84 ^a^	0.28	<0.001
Valerianic acid (%)	4.53 ^a^	3.69 ^b^	3.48 ^b^	3.48 ^b^	3.51 ^b^	3.53 ^b^	0.41	<0.001
A/P ^4^	1.45 ^cd^	1.37 ^e^	1.42 ^de^	1.48 ^c^	1.63 ^b^	1.77 ^a^	0.05	<0.001

^1^ TVFA, Total volatile fatty acids. ^2^ MCP, Microbial protein. ^3^ The unit % represents the proportion of a single VFA in the TVFA. ^4^ A/P, the ratio of acetic acid to propionic acid. SEM: Standard error of the mean. ^a,b,c,d,e^ Values with different superscripts in the same row are significantly different (*p* < 0.05). The same below.

**Table 4 animals-12-00118-t004:** Effects of different additive treatments on diversity index.

Index Type	EOC2	EOC5	CON	SEM	*p*-Valve
Shannon	6.83	6.93	6.77	0.134	0.163
Simpson	0.978	0.979	0.975	0.003	0.17
ACE	558.09	575.31	573.13	14.01	0.057
Chao1	564.1	577.34	576.17	11.26	0.112

## Data Availability

All data for this study are available at the corresponding author. The sequencing data were deposited into the Sequence Read Archive (SRA) of NCBI (Accession Nos. SRR16263676-SRR16263693, https://www.ncbi.nlm.nih.gov/bioproject/PRJNA769507 (accessed on 8 October 2021).

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
