# Peer review of "Effect of Oregano Oil and Cobalt Lactate on Sheep In Vitro Digestibility, Fermentation Characteristics and Rumen Microbial Community"

_animals, 2022, doi:10.3390/ani12010118_

Round 1

Reviewer 1 Report

Effect of oregano essential oil and cobalt lactate combined (EOC) on in vitro fermentation was conducted by used sheep rumen fluid. Gas production, fermentation end-products, and especially microbial population were altered by high level inclusion of EOC. However, some of results are missing interpret including Fig. 1 (no statistical analysis), Table 2 and Table 5 showed no significant by ANOVA (p>0.05), resulted in missing of discussion by author and may be conclusion. Although this manuscript show new and excellent results in microbial community, major revision should be perform.
More detail please see the attached file

Author Response

Dear teacher, thank you for your guidance. I have revised the paper according to your guidance, including statistical analysis of Figure 1, modified Table 2 and Table 5, and modified the content in the corresponding part of the paper.

  1. Line 18: Synergistic effect have to inform to response this objective

Answer: Thank you for your guidance. I have changed this sentence to " Previous studies on EO and Co of oregano showed different results "

  1. Line 21-22: These sentences are contrast, improved mean statistically significant difference. Please rewrite.

Answer: Thank you very much for your review, this sentence has been revised to” the experimental data obtained showed that all five EOC (0.1425% cobalt lactate +1.13% oregano essential oil + 98.7275% carrier) addition rates in this experiment had no significant effect on nutrient digestibility”

  1. Line 21-22: Proportion of EO and Co have to inform or not?

Answer: Thank you very much for your review, Proportion of EO and Co have to inform” EOC (0.1425% cobalt lactate +1.13% oregano essential oil + 98.7275% carrier)”

  1. Line 46: It should not be the optimum level due to this just highest level from the present work, how about if we add EOC more than 1,500 mg/L, it may greater than this or els.

Answer:Thank you very much for your review, you are making a lot of sense, I have deleted this sentence

  1. Line 61: Make sure this information, I think that some compounds still allow with strict requlation

Answer:Thank you very much for your guidance, I have revised it to “has almost banned”

  1. Fig 1: Did this result tested with statistic or not? if did, error bar have to show

Answer:Thank you very much for your help! This result was tested with statistic, and we have added error bars.

  1. Fig 1: The space between time of incubation in graph should not similar, for example space between 12hr-24hr must wider than 2hr-4hr

Answer:Thank you very much, I have adjusted the interval of incubation time in the graph

  1. Table 2: This result have closely discussion, why EOC at low level decrease GP immediately, about 1/3 when compared with CON, while digestion not difference

Answer: Thank you for your guidance, we have added this part to the discussion.

  1. Table 2, Table 5: When ANOVA show not significant, that mean all treatment have similar mean, so no superscript

Answer:Thank you very much for your guidance, I have removed the superscript and explained the possibility of the not significant digestibility in the discussion.

Reviewer 2 Report

Dear authors, it seems to me that this manuscript has great relevance in the scientific world. However, some points could affect the quality of the manuscript.

 Title: Delete “mixture”

 Line 21: What is EOC? Write the acronyms the first time they appear considering the simple summary, the abstract and the body of the text as different parts, where the acronyms must be written the first time that appears.

 Lines 21-22: Dear author, this idea is confusing. The way you say that all the diets improved digestibility, but only one was significant. So how do you know that the other diets improve digestibility? Rewrite it.

 Line 22: What additive, additives or mixture of additives? Rewrite it.

 Line 24: I recommend describing this part more specifically, because it is not correct to generalize about other additives. Rewrite it considering only oregano essential oil and cobalt lactate.

 Line 26: mixture and combined. These words are repetitive. Rewrite it.

 Lines 27-29: Describe the proportion of the additives in the EOC.

 Line 29: Delete “With respect to in vitro fermentation experiments,”

 Lines 36-38: Change “.” Instead “;”

 Lines 44-46: Put these ideas together into one. Consider writing “EOC (1500 mg L-1)”

 Keywords: Delete: culture. Separate oregano and essential oil in two keywords.

 Lines 51-52: If there is a low utilization of by-products, how is there a lack of feed resources? Are you sure? Rewrite it.

 Lines 50-55: There are two ideas here, but these two ideas are not correlated. Rewrite it.

 Line 88: Delete “To determine the exact mechanism of action,” because with the evaluation of these parameters, it is difficult to determine the exact mechanism of action

 Lines 92-93: Delete these lines.

 Line 99: Separate the paragraph after this point.

 Line 100: Add the breed of the animals.

 Line 106: “Fully mixed diet” or “Total mixed diet”?. “Air drying basis” or “Dry matter basis”?

 Table 1: The sum of the ingredient content is not 100. Also, fill in the 2 decimal places for the wheat bran.

 Table 1: Add the neutral detergent fiber, acid detergent fiber, and ether extract contents.

 Line 116: Are you sure, "ratio" is the correct word? Rate - This is a proportion expression, such as the A: B ratio. If not, correct it here and throughout the text.

 Lines 119-122: This text belongs to the results topic, delete or rewrite it.

 Lines 140-147: According to who followed this methodology?

 Lines 149-151: Add the code for these procedures that is in the AOAC book.

 Lines 205-213: Add don’t see the experimental design (Completely randomized design or other?), add this information. Why you use the Duncan test? Are you use covariates?

 Figure 1: Add the p-value when it was significant. Or add a symbol to explain significant points. Also, delete the word “trend”

 Figure 1: Are you sure the IVNDFD and IVADFD values are correct? In theory, the digestibility of IVNDFD should be higher than that of IVADFD.

 Line 227: the right word is “degradability” or “digestibility”

 Figure 4: Describe a better title for this figure.

 The results topic has the same writing style. Rewrite it so it doesn't sound like a collection of repetitive paragraphs.

 Lines 306-308: I understand the idea in these lines, but it is confusing, please rewrite it.

 Line 314: Are you sure? Few reports of essential oils?

 Lines 315-320: Ok, you compared your data with other authors, but remember, the topic of discussion is to biologically explain your results. So what is the biological response to your results?

 Lines 333-337: How this revision can improve your discussion or explain your results?

 Lines 337-343: Only the last two lines are a discussion, although it is generic. Explain your results specifically.

 Lines 344-359: Here I can read a review and comparative data with their results. How do you explain your results? Rewrite it.

 Lines 359-361: How?

 Lines 371-373: This is a text that belongs to the results topic. Delete it from here.

 Lines 371-382: A long but contradictory text. Rewrite it by discussing your data.

 Lines 381-382: How?

 Lines 383-424: Here I can read a review, data of the results topic, and comparative data with their results. How do you explain your results? Rewrite it.

 Lines 425-442: Only 434-436 lines are a discussion in this paragraph. Rewrite it as a discussion. E.g.: Why microorganisms are positively or negatively correlated.

 Line 444: You didn't test the EOC concentrations, you tested the increasing levels of EOC inclusion in the diet.

 Line 444: Delete “cultures”

 Conclusion: Lines 446-448 could be first than lines 444-446. Also, delete lines 448-449 or combine them with lines 446-448, because this way it is repetitive.

 Organize the material and methods topic in chronological or sequential order.

Author Response

Thank you for your guidance. I have revised the paper comprehensively under your guidance. Thank you very much for your help

  1. Title: Delete “mixture”

Answer:I have deleted “mixture”

  1. Line 21: What is EOC? Write the acronyms the first time they appear considering the simple summary, the abstract and the body of the text as different parts, where the acronyms must be written the first time that appears.

Answer:Thank you for your guidance, I have revised this sentence to "EOC (0.1425% cobalt lactate +1.13% oregano essential oil + 98.7275% carrier)".

  1. Lines 21-22: Dear author, this idea is confusing. The way you say that all the diets improved digestibility, but only one was significant. So how do you know that the other diets improve digestibility? Rewrite it.

Answer:Respected reviewers, one of them noted “when ANOVA show not significant, that mean all treatment have similar mean, so no superscript”, we have thus revised it to have no significant effect on nutrient digestibility.

  1. Line 22: What additive, additives or mixture of additives? Rewrite it.

Answer:I have revised to “the addition of 1500 mg·L-1 EOC”.

  1. Line 24: I recommend describing this part more specifically, because it is not correct to generalize about other additives. Rewrite it considering only oregano essential oil and cobalt lactate.

Answer:I have revised to “All presented data provide a theoretical basis for the application of oregano essential oil and cobalt in ruminant nutrition.”

  1. Line 26: mixture and combined. These words are repetitive. Rewrite it.

Answer: I have revised to “The objective of this experiment was to evaluate the effect of different EOC (0.1425% cobalt lactate +1.13% oregano essential oil + 98.7275% carrier) levels on in vitro rumen fermentation and microbial changes”

  1. Lines 27-29: Describe the proportion of the additives in the EOC.

Answer: I have been describe the proportion of the additives in the EOC (0.1425% cobalt lactate +1.13% oregano essential oil + 98.7275% carrier).

  1. Line 29: Delete “With respect to in vitro fermentation experiments,”

Answer: This sentence has been deleted

  1. Lines 36-38: Change “.” Instead “;”

Answer: I have been Change “.” Instead “;”

  1. Lines 44-46: Put these ideas together into one. Consider writing “EOC (1500 mg L-1)”

Answer: Thanks for your guidance, I have revised this sentence to “Comprehensive analysis showed that EOC (1500 mg·L-1) could significantly increase gas production, alter sheep rumen fermentation parameters and microbiota composition.”.

  1. Keywords: Delete: culture. Separate oregano and essential oil in two keywords.

Answer: has been deleted: culture, And separate oregano and essential oil in two keywords.

  1. Lines 51-52: If there is a low utilization of by-products, how is there a lack of feed resources? Are you sure? Rewrite it.

Answer: Thank you very much for the expert guidance, I have modified it

  1. Lines 50-55: There are two ideas here, but these two ideas are not correlated. Rewrite it.

Answer: Thank you very much for your help, I have modified it to “The exploitation and utilization of straw feed has always been an important subject in animal nutrition, massive burning leads to serious waste of crop straw, although it is difficult to be applied on monogastric animals[1], but the microbial community of ruminants allows them to produce high-quality protein from low-quality plant feeds and catalyze the hydrolysis of indigestible plant crude fibers through microbial extracellular enzymes”

  1. Line 88: Delete “To determine the exact mechanism of action,” because with the evaluation of these parameters, it is difficult to determine the exact mechanism of action

Answer: Thank you very much for your guidance, I have deleted it

  1. Lines 92-93: Delete these lines.

Answer: Thank you very much for your guidance, I have deleted it

  1. Line 99: Separate the paragraph after this point.

Answer: Thank you very much for your guidance, I have Separate the paragraph after this point.

  1. Line 100: Add the breed of the animals.

Answer: Thank you very much for your guidance, I have add the breed of the animals” small-tailed han sheep”

  1. Line 106: “Fully mixed diet” or “Total mixed diet”?. “Air drying basis” or “Dry matter basis”?

Answer: Answer: Thank you very much for your guidance! This is “Total mixed diet”, I have corrected it in the manuscript. Air-drying basis is right, the dry matter content of it is 36.23.

  1. Table 1: The sum of the ingredient content is not 100. Also, fill in the 2 decimal places for the wheat bran.

Answer: Thank you very much for checking the data, I rechecked the data and corrected it and fill in the 2 decimal places for the wheat bran.

  1. Table 1: Add the neutral detergent fiber, acid detergent fiber, and ether extract contents.

Answer:respected teacher. I have Add the neutral detergent fiber, acid detergent fiber, and ether extract contents.

  1. Line 116: Are you sure, "ratio" is the correct word? Rate - This is a proportion expression, Answer:such as the A: B ratio. If not, correct it here and throughout the text.

Thank you very much for your guidance! I have changed "inclusion ratio" to "inclusion levels" throughout the text.

  1. Lines 119-122: This text belongs to the results topic, delete or rewrite it.

Answer: Thank you very much for your guidance! I have delete it

  1. Lines 140-147: According to who followed this methodology?

Answer: Thank you for your guidance, I have added the IVDMD measurement guidelines and the VFA, NH3-N, and MCP measurement methods described in the next paragraph.

  1. Lines 149-151: Add the code for these procedures that is in the AOAC book.

Answer: The in vitro dry matter digestibility (IVDMD) was determined by drying at 65 °C for 48 h (AOAC, 2016; 930.15), in vitro neutral detergent fiber (AOAC, 2016; 973.18) and in vitro acid detergent fiber (AOAC, 2016; 2002.04) using the AOAC international procedures

  1. Lines 205-213: Add don’t see the experimental design (Completely randomized design or other?), add this information. Why you use the Duncan test? Are you use covariates?

Answer: We used a one-way experimental design, Duncan's method is a classical multi-field comparison method after one-way anOVA, we used SPSS (SPSS v 26.0, SPSS, Inc.)'s Kolmogorov-Smirnov test program to check the normality and outliers of all data. Levene's test was used to test the homogeneity of variance, All of the above values were normal, so we did not perform covariate analysis

  1. Figure 1: Add the p-value when it was significant. Or add a symbol to explain significant points. Also, delete the word “trend”

Answer: Thank you very much for your guidance! I have add “*”to explain significant points, and delete the word “trend”

  1. Figure 1: Are you sure the IVNDFD and IVADFD values are correct? In theory, the digestibility of IVNDFD should be higher than that of IVADFD.

Answer: Thank you very much for your correction of the data, we found that the formula for nutrient digestibility was used incorrectly after checking. It has now been corrected to “nutrient digestibility (%) = 100 × (sample nutrient concentration - residue nutrient concentration)/sample nutrient concentration”. We have modified Table 2 and changed the description accordingly in the whole text.

  1. Line 227: the right word is “degradability” or “digestibility”

Answer: I have changed the word to“digestibility”

  1. Figure 4: Describe a better title for this figure.

Answer: Thank you for your suggestion, the title has been revised to “ Significantly different bacterial taxa identified by the linear discriminant analysis effect size (LEfSe)”.

  1. The results topic has the same writing style. Rewrite it so it doesn't sound like a collection of repetitive paragraphs.

Answer: Thank you for your guidance! The titles of sections 3.1-3.4 have been revised.

  1. Lines 306-308: I understand the idea in these lines, but it is confusing, please rewrite it.

Answer:I have been revised to“Previous studies have shown different effects of oregano essential oil and cobalt on rumen microbial activity and fermentation characteristics in ruminants due to different types and addition levels of EO”.

  1. Line 314: Are you sure? Few reports of essential oils?

Answer: Thank you for your suggestion, the sentence has been removed.

  1. Lines 315-320: Ok, you compared your data with other authors, but remember, the topic of discussion is to biologically explain your results. So what is the biological response to your results?

Answer:Thank you very much for your guidance, it has benefited me a lot and I have rewritten the discussion section according to your instructions.

  1. Lines 333-337: How this revision can improve your discussion or explain your results?

Answer: Thank you very much for your guidance, I have rewritten the discussion section in terms of explaining my results

  1. Lines 337-343: Only the last two lines are a discussion, although it is generic. Explain your results specifically.

Answer:Thank you very much for your guidance, it has benefited me a lot and I have rewritten the discussion section according to your instructions.

  1. Lines 344-359: Here I can read a review and comparative data with their results. How do you explain your results? Rewrite it.

Answer: Thank you very much for your guidance, it makes me understand how to discuss

  1. Lines 371-373: This is a text that belongs to the results topic. Delete it from here.

Answer: Thank you very much for your guidance, I have deleted this sentence.

  1. Lines 371-382: A long but contradictory text. Rewrite it by discussing your data.

Answer: I have rewritten the sentence

  1. Lines 381-382: How?

Answer: We have described in the discussion section the pathway of Co for improving rumen fermentation

  1. Lines 383-424: Here I can read a review, data of the results topic, and comparative data with their results. How do you explain your results? Rewrite it.

Answer: Thanks for your guidance, we have rewritten the entire discussion section by explaining our data

  1. Lines 425-442: Only 434-436 lines are a discussion in this paragraph. Rewrite it as a discussion. E.g.: Why microorganisms are positively or negatively correlated.

Answer: Thanks for your guidance, the correlation between rumen microorganisms and rumen fermentation levels was found in this experiment by correlation analysis and we have discussed some of the reasons for it in the microbiology section.

  1. Line 444: You didn't test the EOC concentrations, you tested the increasing levels of EOC inclusion in the diet.

Answer: I have changed "concentrations" to “inclusion levels”

  1. Line 444: Delete “cultures”

Answer: Thank you for your guidance. I have been delete “cultures”

  1. Conclusion: Lines 446-448 could be first than lines 444-446. Also, delete lines 448-449 or Answer: combine them with lines 446-448, because this way it is repetitive.

We have placed lines 446-448 before lines 444-446 and deleted lines 448-449

  1. Organize the material and methods topic in chronological or sequential order.

Answer: I had Reorganized material and method topics chronologically

Reviewer 3 Report

The manuscript submitted by Wang et al. deals with the evaluation of the effect of oregano oil and cobalt lactate mixture on sheep in vitro digestibility, fermentation characteristics and rumen microbial community. The paperi s interesting and quite original, the experimental design is clear and no ethical issues need to be addressed since authors provided approval  by the Animal Committee of Gansu Agricultural University.

Specific points to address:

  • It would be useful if the authors managed to streamline the abstract.
  • L88-93: The objective of the study is clearly presented, however the hypothesis on the basis of which the authors designed the study must be better indicated.
  • L131: remove the round bracket.
  • The methods applied must be better described. In particular, more information is needed about the DNA extraction (L172-174).
  • The discussion is orderly and clear. The only major limitation concerns the fact that for how the study was designed, it is not possible to state whether a given effect is more attributable to organ essential oil or to cobalt lactate, as all treatments involved the combined use of these elements. The discussion must be reviewed with this aspect in mind.
  • The limitations highlighted for discussion must be also indicated in the conclusions section.

Author Response

Thank you very much for your guidance. I have revised the paper with your help.

     1. It would be useful if the authors managed to streamline the abstract.

Answer: Thank you for your help. I have simplified the summary

     2. L88-93: The objective of the study is clearly presented, however the hypothesis on the basis of which the authors designed the study must be better indicated.

Answer: Thank you for your help. I've added “This study hypothesized that different levels of EOC addition would have different effects on rumen microorganisms, while nutrient digestibility and fermentation parameters might change with the microbiota altered. Therefore,”

      3. L131: remove the round bracket.

Answer:I have removed the round bracket.

      4. The methods applied must be better described. In particular, more information is needed about the DNA extraction (L172-174).

Answer: Thank you for your guidance, we have added the detailed steps for DNA extraction

     5. The discussion is orderly and clear. The only major limitation concerns the fact that for how the study was designed, it is not possible to state whether a given effect is more attributable to organ essential oil or to cobalt lactate, as all treatments involved the combined use of these elements. The discussion must be reviewed with this aspect in mind. The limitations highlighted for discussion must be also indicated in the conclusions section.

Answer:Thank you for your guidance. We have revised the discussion section again in light of your guidance and have noted the salient limitations for discussion in the conclusion section.

Round 2

Reviewer 1 Report

Accept in present form

Reviewer 2 Report

Dear authors, I congratulate you on the work you have done.  You made corrections to the manuscript according to my suggestions.

Reviewer 3 Report

Authors addressed all concerns raised during the revision process.